# The Mediator Complex: A Central Coordinator of Plant Adaptive Responses to Environmental Stresses

**DOI:** 10.3390/ijms23116170

**Published:** 2022-05-31

**Authors:** Jialuo Chen, Su Yang, Baofang Fan, Cheng Zhu, Zhixiang Chen

**Affiliations:** 1College of Life Sciences, China Jiliang University, Hangzhou 310018, China; s20090710003@cjlu.edu.cn (J.C.); yangsu@cjlu.edu.cn (S.Y.); 2Department of Botany and Plant Pathology, Purdue University, West Lafayette, IN 47907, USA; bfan@purdue.edu

**Keywords:** mediator complex, plant stress responses, transcriptional regulation, biotic and abiotic stress

## Abstract

As sessile organisms, plants are constantly exposed to a variety of environmental stresses and have evolved adaptive mechanisms, including transcriptional reprogramming, in order to survive or acclimate under adverse conditions. Over the past several decades, a large number of gene-specific transcription factors have been identified in the transcriptional regulation of plant adaptive responses. The Mediator complex plays a key role in transducing signals from gene-specific transcription factors to the transcription machinery to activate or repress target gene expression. Since its first purification about 15 years ago, plant Mediator complex has been extensively analyzed for its composition and biological functions. Mutants of many plant Mediator subunits are not lethal but are compromised in growth, development and response to biotic and abiotic stress, underscoring a particularly important role in plant adaptive responses. Plant Mediator subunits also interact with partners other than transcription factors and components of the transcription machinery, indicating the complexity of the regulation of gene expression by plant Mediator complex. Here, we present a comprehensive discussion of recent analyses of the structure and function of plant Mediator complex, with a particular focus on its roles in plant adaptive responses to a wide spectrum of environmental stresses and associated biological processes.

## 1. Introduction

Transcription is the first step in gene expression. In eukaryotes, all nuclear protein-coding genes are transcribed by RNA polymerase II (Pol II). Pol II and several general transcription factors (GTFs), including transcription initiation factor IIA (TFIIA), TFIIB, TFIID, TFIIE, TFIIF and TFIIH, assemble on the core promoters, forming a large multiprotein-DNA preinitiation complex (PIC) for accurate transcription initiation [1,2]. Gene-specific transcription factors determine the tissue- and cell-specific rates of gene expression [3]. These specific transcription factors recognize specific upstream cis-acting DNA elements of their target genes and activate transcription through recruitment of coactivators [4]. Some coactivators act as chromatin modifiers or remodelers to mobilize nucleosomes, the basic unit of chromatin structure, to facilitate access of the transcription apparatus and other regulators to DNA [5]. Gene-specific transcription factors can also recruit histone-modifying enzymes to chemically modify nucleosomes across active genes to provide surfaces for interaction with regulatory protein complexes of transcription [4]. Other coactivators or regulators, such as the Mediator complex, play an important role in transmitting information from gene-specific transcription factors to the transcription machinery assembled at promoters as PIC to control transcription initiation [6,7] (Figure 1).

Mediator is a large complex with 25–40 subunits that are organized into three core modules (head, middle and tail) and the dissociable cyclin-dependent kinase 8 (CDK8) module (CKM) [7,8]. In the budding yeast, the head, middle and tail modules, as well as CKM, consist of seven (MED6, MED8, MED11, MED17, MED18, MED20 and MED22), eight (MED1, MED4, MED7, MED9, MED10, MED19, MED21 and MED31), five (MED2, MED3, MED5, MED15 and MED16) and four (MED12, MED13, CDK8 and C-type cyclin or CycC) subunits, respectively [9]. MED14 makes contact with the head, middle and tail modules of the Mediator complex and is a central backbone for the three main Mediator modules, whereas CKM is only transiently associated with the Mediator complex [7]. Five additional subunits (MED23, MED25, MED26, MED28 and MED30) and several paralog components of CKM are also present in mammalian Mediator complex [7]. The Mediator complex is recruited to enhancer regions by specific transcription factors, which often involves direct interactions between specific transcription factors and Mediator tail module subunits [7,10] (Figure 1). Then, the Mediator complex can cooperate with PIC components, such as TFIIB, TFIID, TFIIH and Pol II, to promote PIC assembly through specific protein–protein interactions involving both the Mediator head and middle modules [7] (Figure 1). CKM generally functions as a transcription repressor under steady-state growth conditions [11]. The binding of CKM to the core Mediator complex sterically inhibits Mediator from binding to PIC, and the activity of Mediator to stimulate transcription initiation is accompanied by the dissociation of CKM [11]. However, under certain stress conditions, CKM can function as a positive regulator of gene transcription, possibly by releasing Mediator and allowing for its interaction with PIC [11].

Since it was first purified from Arabidopsis more than 15 years ago [12], plant Mediator complex has been extensively analyzed. First, a combination of biochemical, molecular and bioinformatics approaches led to successful identification and annotation of not only conserved but also divergent and plant-specific Mediator subunits in Arabidopsis and other plants [13] (Table 1). Second, mutants for many Arabidopsis Mediator subunits have been isolated and functionally characterized. Unlike in yeast and mammalians, where knockout of various Mediator subunits is lethal, mutations in many Arabidopsis Mediator subunits are not lethal but cause defects in growth, development and response to biotic and abiotic stimuli [13,14,15,16,17]. These observations suggest that plant Mediator complex has a particularly important role in plant adaptive responses and is an ideal system for analysis of the mechanisms by which Mediator regulates transcription of target genes and associated biological processes (Table 1). In this review, we discuss recent developments in the investigation of the structure and function of plant Mediator, with particular focus on its roles in the regulation of plant adaptive responses to a wide spectrum of environmental stresses and associated biological processes (Table 1).

## 2. Conserved and Unique Features of Transcription and Mediator Complex in Plants

As sessile organisms, plants have many unique traits in terms of growth, development, metabolism and responses to environmental stimuli. Although the basic mechanisms of transcription are highly conserved in all eukaryotes, plants have distinct programs for transcriptional regulation of gene expression. A case study was conducted on special transcription control in plants in the presence of two non-redundant plant-specific RNA polymerases, Pol IV and Pol V, which produce non-coding RNAs required for transcriptional gene silencing through the RNA-directed DNA methylation (RdDM) pathway [18]. In addition, expansion of general transcription factors has been observed, particularly the TFIIB-like factors, with at least 14 members in Arabidopsis [19,20]. In contrast, TFIIB often has only two homologs, Rrn7 and Brf1, in other eukaryotes as general transcription factors for Pol I and Pol III, respectively [19]. Molecular and genetic analysis has revealed important functions of these TFIIB-like factors in many plant-specific biological processes, including gametogenesis, pollen tube growth guidance, endosperm development and plant–microbe interactions [20]. There is a functional interplay between Mediator and TFIIB in PIC assembly, which is closely related to the promoter architecture through specific protein–protein interactions [21]. The expansion of the TFIIB family in plants could have important implications in the formation of multiple PICs and their physical and functional interactions with Mediator in the transcription regulation of plant genes.

Although the overall Mediator structure and function are well conserved among yeast, animals and plants, numerous studies have revealed significant divergence in plant Mediator composition and its coordination with other components of transcription machinery. Purification of plant Mediator from Arabidopsis suspension cultures was first reported in 2007 [12]. Mass spectrometry of purified Arabidopsis Mediator identified 19 conserved MED subunits associated with the head, middle and tail modules. CKM was not copurified in the purified Arabidopsis Mediator, although its subunits (MED12, MED13, CDK8 and CycC) have already been identified from genetic and sequence analysis [22,23]. There were also six plant-specific MED subunits (MED32-MED37) in the purified Arabidopsis Mediator, but subsequent sequence analysis indicated that Arabidopsis MED32 and MED33 are orthologs of human MED29 and MED24 and yeast MED2 and MED5, respectively [24] (Table 1). Among the remaining four plant-specific Mediator subunits, MED34 is a (d)NTP-dependent 3′ -> 5′ DNA helicase (RecQ-like2 or RECQL2). MED35 is an mRNA-splicing factor (PRE-mRNA PROCESSING PROTEIN 40A or PRP40A). MED36 is an rRNA methyltransferase (FIBRILLARIN 2 or FIB2) that can methylate histone H2A in the promoter of ribosomal genes. MED37 is an endoplasmic reticulum (ER)-localized heat shock protein 70 (BINDING PROTEIN1 or BIP1).

In a more recent study, Guo and colleagues reported affinity purification of Arabidopsis Mediator from transgenic Arabidopsis plants expressing six FLAG-tagged conserved Mediator subunits: MED8, MED11, MED18, MED4, MED31 and MED25 [25]. Mass spectrometry identified 28 conserve Mediator subunits, including those of CKM (MED12, MED13, CDK8 CycC1-1 and CycC1-2) [25]. These results provide experimental evidence that CKM is part of the Arabidopsis Mediator complex. In contrast to the Arabidopsis Mediator purified from suspension cultures [12], the purified Mediator complex from transgenic Arabidopsis plants contains no MED34, MED35 or MED36 [25]. MED37A was identified not only in multiple Mediator transgenic plants but also in wild-type (WT) control plants [25]. These results do not support the classification of MED34-MED37 as plant-specific Mediator subunits. On the other hand, two homologs of cyclic adenosine monophosphate response element binding protein (CREB) binding protein (CBP)/p300 histone acetyltransferases, HAC1 and HAC5, were copurified with the Mediator complex from transgenic Arabidopsis plants (Table 1). Further analysis indicated that Mediator subunits MED8 and MED25 are partially responsible for the association of Mediator with HAC1 and HAC5 [25]. Arabidopsis MED8/25/ and HAC1/5 coregulate gene expression to affect flowering time and floral development [25]. HAC1 also interacts with MED25 in the regulation of gene transcription during jasmonic acid (JA) signaling [26]. The copurification of HAC1 and HAC5 with Arabidopsis Mediator indicates that these CBP/p300 proteins interact with the whole Mediator complex and likely function as plant-specific Mediator subunits. Intriguingly, there is also synergism between CBP/p300 and Mediator in the regulation of transcription in mammals [27,28].

Although Mediator is generally considered to be a coactivator complex for Pol II, there have been reports that yeast Mediator is involved in regulating Pol III transcription of tRNAs [29]. In support of a broad role of Mediator in transcription, it has been reported that yeast Mediator is associated not only with Pol I and Pol III but also with proteins involved in transcription elongation and RNA processing [30]. In Arabidopsis, Mediator subunit MED18, which regulates flowering, hormone signaling and plant immunity [31,32], also interacts with NUCLEAR RNA POLYMERASE D2a (NRPD2a) [33], the second largest subunit of plant-specific Pol IV and Pol V [33]. Mutants for both MED18 and NRPD2a share phenotypes in compromised resistance to necrotrophic fungal pathogen *Botrytis cinerea*, reduced accumulation of reactive oxygen species, altered levels of salicylic acid (SA) and JA and expression of genes associated with plant–pathogen interactions [33]. The novel physical and functional interaction between subunits of Mediator and plant-specific RNA Pol IV and Pol V provides strong evidence for a critical role of Mediator in RNA-directed DNA methylation and epigenetic regulation of plant genes.

## 3. Mediator Complex in Signaling of JA, a Stress-Responsive Hormone

JA is a lipid-derived plant hormone that regulates plant responses to both biotic and abiotc stresses by activating genome-wide transcription of JA-responsive genes [34,35]. The basic helix–loop–helix (bHLH) transcription factor MYC2 is a master regulator of JA signaling and responses [36,37]. In the absence of jasmonoyl-isoleucine (JA-Ile), the bioactive JA ligand, MYC2 is associated with a group of JASMONATE-ZIM DOMAIN (JAZ) repressor proteins, which recruit the general co-repressor TOPLESS (TPL) to form a repressor complex that prevents MYC2 from activating transcription of JA-responsive genes [38,39,40]. In the presence of JA-Ile, JAZ proteins form complexes with the JA receptor CORONATINE-INSENSITIVE1 (COI1), which is an F-box subunit of SCF^COI1^ ubiquitin E3 ligase and targeted for degradation by the 26S proteasome [41,42,43]. Degradation of JAZ proteins leads to derepression of MYC2 to activate the transcription of JA-responsive genes.

The Mediator complex plays an essential role in diverse aspects of JA signaling and responses. The transcriptional repressive activity of the TPL general corepressor is directly associated with the Mediator complex [44]. TPL is recruited by numerous transcription repressors, including JAZ and Aux/IAA proteins, to repress JA- and auxin-responsive genes, respectively [45,46,47]. TPL also recruits CKM to the Mediator complex through direct interaction with the MED13 subunit of CKM, thereby inhibiting Mediator from binding to PIC and the activity of Mediator to stimulate transcription [48]. In the presence of high auxin levels, proteolysis of Aux/IAA repressor proteins releases TPL and CKM, leading to derepression of both ARF transcription activators and Mediator to activate auxin-responsive genes [48]. A similar mechanism for derepression of MYC2 is also likely in the presence of high JA-Ile to activate JA-responsive genes. More recently, it has been reported that the potent repression domain in helix 8 within the CT11-RanBPM (CRA) domain of TPL also directly interacts with the Mediator middle module subunits, MED21 and MED10 [44]. The interaction between TPL and the Mediator subunits, apparently for recruitment of the entire core Mediator complex (head, middle and tail) to TPL-repressed loci, is required to maintain repression [44]. Further analysis suggests that TPL binding prevents the formation of a fully active Mediator complex that is unable to recruit Pol II for assembly of PIC. TPL recognizes the N-terminus of MED21, which is required for recruitment of Pol II [44]. Thus, TPL can regulate Mediator activity through at least two different mechanisms by binding to distinct Mediator subunits.

The multifunctional Mediator subunit MED25 acts an integrator of JA signaling by physically interacting not only with components of JA signaling but also with multiple transcription and epigenetic regulators [49]. First, MED25 interacts with JA receptor COI1 and some JAZ repressors and promotes COI1–JAZ interaction and subsequent JAZ degradation in response to active JA-Ile hormones [26] (Figure 2). JAZ degradation leads to weakened MED25–COI1 interaction. Second, upon JA-dependent JAZ degradation, MYC2 recruits MED25 to the promoters of MYC2-targeted JA-responsive genes through direct interaction to promote PIC assembly for their transcriptional activation [50] (Figure 2). MYC2-targeted JA-responsive genes also include those encoding intermediate transcription factors, such as AP2/ERF transcription factors ORA59 and ERF1, which further recruit MED25 through physical interaction to activate downstream JA-responsive defense-related genes, thereby amplifying JA-mediated transcriptional output [51,52]. Third, MED25 interacts with histone acetyltransferases HAC1 and HAC5 on MYC2 target gene promoters (Figure 2). The HAC1 and HAC5 epigenetic regulators selectively regulate histone H3 lysine 9 (H3K9) acetylation at MYC2 target gene promoters to promote Pol II recruitment and transcription [26]. The MYC2–MED25–HAC1 interactions are enhanced in the presence of JA, largely due to the action of the Groucho/Thymidine uptake 1(Gro/Tup1) family protein LEUNIG_HOMOLOG (LUH) [53]. LUH promotes MYC2–MED25–HAC1 interactions by interacting with both MED25 and HAC1 through distinct domains to amplify the transcription output in JA signaling [53] (Figure 2). Fourth, MED25 is involved in finetuning of transcription output of JA signaling to prevent excessive activation of JA-mediated defense responses, which could lead to severe growth inhibition. One mechanism for finetuning is through regulation of alternative splicing of JAZ genes, which generates JAZ repressor variants that are unable to interact with SCF^COI1^ for degradation [54,55] (Figure 2). Both MYC2 and MED25 are required for JA-induced production of JAZ splicing variants [56]. On the other hand, MED25 also physically interacts with splicing factors PRP39a and PRP40a to recruit them to JAZ loci to promote the full splicing of JAZ genes, thereby maintaining JAZ variants at appropriate levels [56]. Another mechanism for finetuning is through induction of negative regulators of JA-mediated transcriptional responses. In tomato, MYC2 and MED25 activate expression of a group of MYC2-targeted BHLH (MTB) proteins, which contain DNA-binding and JAZ-interacting domains, like MYC2, but lack the MED25-interacting domain [57]. As a result, these MTB proteins function as JA-induced transcription repressors by disrupting the interaction of MYC2 with MED25 and by impairing the binding of MYC2 to its target promoters [57] (Figure 2). Therefore, MED25 plays an important role in almost every step of JA-mediated transcription, including SCFCOI1-mediated degradation of JAZ repressors, MYC2-dependent PIC assembly, epigenetic regulation, mRNA processing and induction of MTB repressors for negative feedback regulation of JA signaling.

## 4. Function of the Mediator Complex in Broad Plant Biotic Interactions

Plants respond to pathogens using two interconnected innate immune systems: PTI (pathogen-associated molecular pattern (PAMP)-triggered immunity) and ETI (effector-triggered immunity) [58]. PTI is activated by PAMPs, such as bacterial flagellin upon their recognition by cell surface pattern-recognizing receptors (PRRs). To suppress PTI, pathogens deliver effectors to plant cells to promote infection. However, some of the effectors can activate ETI upon their recognition, either directly or indirectly, by plant disease resistance proteins. ETI is often manifested as hypersensitive responses (HRs), which are associated with accumulation of plant defense signal molecule salicylic acid (SA) not only in infected cells but also in upper uninfected tissues to establish systemic acquired resistance (SAR) [59]. HR and SA-mediated SAR are important defense mechanisms against biotrophic pathogens, which rely on a feeding relationship with the living plant cells during the infection process. Studies conducted in the past decade have shown that plant Mediator is closely involved in the regulation of various aspects of plant innate immune systems (Figure 3).

One of the best characterized bacterial PAMPs is elf18, which is derived from bacterial translation elongation factor EF-Tu and recognized by the EF-Tu receptor (EFR) in Arabidopsis [60]. Previously, an ELF18-induced long non-coding RNA, ELENA1, was identified and shown to be a positive transcriptional regulator of immune-responsive genes [61]. ELENA1 directly interacts with MED19a and promotes enrichment of the complex at the *PATHOGENESIS-RELATED GENE 1*(*PR1*) promoter to enhance the expression of the defense gene (Figure 3). More recently, it was shown that FIB2(MED36a) can directly interact with both ELENA1 and MED19a [62] (Figure 3). However, unlike ELENA1 and MED19a, FIB2(MED36a) functions as a negative transcriptional regulator of for immune-responsive genes, including *PR1* (Figure 3). Further genetic and biochemical investigation has revealed that the FIB2(MED36a) repressor suppresses the transcription-activating activity of MED19a at the *PR1* promoter to repress its expression [62] (Figure 3). ELENA1 can evict FIB2(MED36a) from the MED19a activator to derepress *PR1* gene expression [62] (Figure 3). These studies reveal complexity in the transcriptional regulation of defense-related genes involving not only transcription factors, Mediator subunits and long non-coding RNA. The critical role of MED19a in plant defense is supported by the finding that this Mediator subunit is targeted for degradation by the effector protein HaRxL44 from oomycete downy mildew pathogen *Hyaloperonospora arabidopsidis* [63] (Figure 3).

Genetic analysis has shown that CKM functions as a positive regulator of SA accumulation and SAR [64] (Figure 3). In Arabidopsis, calmodulin-binding transcription activators CAMTA1, 2 and 3 function redundantly as negative regulators of plant immunity, and their triple mutants display severe dwarfism due to autoimmunity [64] (Figure 3). Genetic suppressor screens led to the discovery that a mutation of *CDK8* gene could partially suppress the dwarf, constitutive SA accumulation and resistance phenotypes of the *camta1/2/3* triple mutant [64]. In the *cdk8* mutants, both free and total SA levels were reduced under uninfected conditions but returned to normal levels 24 h after pathogen inoculation relative to those in WT. These results indicate that CDK8 regulates steady-state SA accumulation. Likewise, expression of SA biosynthetic *ICS1* and *EDS5* genes was downregulated under uninfected conditions but returned to normal after pathogen inoculation in the *cdk8* mutants when compared to those in WT (Figure 3). Importantly, the *cdk8* mutants were compromised in SAR [64]. Mutations of the gene for the MED12 subunit of CKM resulted in similar phenotypes of reduced steady-state SA levels and compromised SAR [64]. These results indicate that the whole CKM of plant Mediator plays a critical role in the transcriptional regulation of SA accumulation and SAR (Figure 3).

NONEXPRESSOR OF PR1 (NPR1) is required for SA-induced defense responses and reprogramming of large-scale gene expression associated with SAR [65]. Recombinant NPR1 binds SA [66,67,68], and this SA-binding activity is required for the activation of SA-responsive defense genes by NPR1 [66]. Upon SAR induction, NPR1 is localized to the nucleus, where it interacts with TGA and TCP transcription factors to activate *PR* gene expression [69,70]. NPR1 is expressed at low levels in healthy uninfected plants and can be induced by pathogen infection and SA treatment. Induction of NPR1 may be a critical step in plant defense responses. We previously identified W-box sequences in the *NPR1* gene promoter that are recognized by SA-induced WRKY transcription factors, such as WRKY18 from Arabidopsis [71]. We showed that SA-induced WRKY transcription factors act upstream of NPR1 and positively regulate its expression during the activation of plant defense responses [71]. More recently, it was reported that SA promotes NPR1 interaction with both CDK8 and WRKY18 to recruit them to its own promoter and enhance its own expression [72] (Figure 3). Consistent with this finding, CDK8 and its associated MED12 and MED13 positively regulate *NPR1* and *PR1* expression and play an important role in both local defense and SAR. CDK8 also interacts with WRKY6, WRKY18 and TGA transcription factors and recruits Pol II to the *NPR1* and *PR1* loci promoters to promote their expression (Figure 3). These results indicate that CKM of plant Mediator plays a critical role in transmitting information from NPR1, WRKY and TGA transcription activators to the transcription machinery to facilitate transcription of *NPR1* and *PR1* genes during plant defense responses [72] (Figure 3).

Contrary to biotrophic pathogens, necrotrophic pathogens kill plant cells at very early stages of infection to extract nutrients from dead or dying host cells [73]. HR, which is highly effective against biotrophic pathogens, is promoted by necrotrophic pathogens to facilitate infection [74]. In addition, resistance to biotrophic pathogens is associated with enhanced SA signaling, resistance to necrotrophic pathogens depends on JA and ethylene signaling and the synthesis of the phytoalexin camalexin [75,76,77,78,79]. Given its positive role in JA signaling, MED25 is required for plant defense against necrotrophic pathogens. Mutation of *MED25* in Arabidopsis increases susceptibility to necrotrophic pathogens *Alternaria brassicicola*, *Botrytis cinerea* and *Sclerotinia sclerotiorum* [80,81]. MED8 can act together with MED25 in regulation of plant immunity or act independently through interaction with a bHLH transcription factor, FAMA, which also plays a critical role in plant resistance to *B. cinerea* [82]. FAMA also interacts with some JAZ proteins and regulates JA-mediated defense genes, including THIOGLUCOSIDASE GLUCOHYDROLASE 1 (TGG1), which encodes a myrosinase that hydrolyzes glucosinolates to produce chemically active compounds that are highly toxic to herbivores and pathogens [83].

MED18 also plays a critical role in defense against necrotrophic pathogens. Arabidopsis *med18* mutants displayed enhanced susceptibility to *B. cinerea* but with increased expression of *PDF1.2*, a JA/ethylene-responsive gene [31]. This result indicates that unlike MED25, MED18 appears to regulate plant immune response against necrotrophic pathogens through a JA-independent pathway. MED18 interacts with a zinc-finger transcription factor, YIN YANG 1(YY1), the mutations of which also result in increased susceptibility to *B. cinereal* [31]. MED18 and YY1 suppress expression of glutaredoxin genes *GRX480* and *GRXS13*, as well as thioredoxin gene *TRX-h5*, which promote disease susceptibility to *B. cinereal* [31]. MED18 also interacts with HOOKLESS1 (HLS1), a putative histone acetyltransferase, to promote the transcription of *WRKY33*, which encodes a transcription factor important for pathogen-induced phytoalexin biosynthesis and resistance to necrotrophic pathogens [32,84,85]. As discussed earlier, MED18 also interacts with NRPD2a, the second largest subunit of plant-specific nuclear Pol IV and Pol V [33]. Mutants for both MED18 and NRPD2a were found to be hypersusceptible to necrotrophic fungal pathogen *B. cinerea*. Mutants for NRPD1a, the largest subunit of Pol IV, also displayed enhanced susceptibility to the necrotrophic pathogen. Transcriptome profiling identified altered expression of genes in the *med18* and *nrpd2a* mutants associated with disease-resistance proteins, as well as SA and JA signaling and responses [33]. The interaction between MED18 and plant-specific RNA polymerases provides a new mechanism for epigenetic regulation in plant immunity.

Other Mediator subunits with critical roles in plant defense against necrotrophic pathogens include MED21 and CDK8. Arabidopsis RNA interference lines with reduced *MED21* expression are highly susceptible to *A. brassicicola* and *B. cinereal* [86]. MED21 interacts with HISTONE MONOUBIQUITINATION1 (HUB1), a RING E3 ligase. Mutants of *HUB1* also display increased susceptibility to the necrotrophic fungal pathogens, whereas HUB1 overexpression enhances resistance to *B. cinereal* [86]. The role of HUB1 in defense against necrotrophic pathogens is independent of JA and subject to the influence of ethylene and SA [86]. The MED21–HUB1 interaction is likely important for chromatin modification at specific defense gene loci and plays regulatory roles in plant defense against necrotrophic fungi through modulation of gene expression [86]. The roles of CDK8 in plant resistance to necrotrophic pathogens are somewhat more complicated [87]. First, Arabidopsis *cdk8* mutants exhibit enhanced susceptibility to *A. brassicicola* but resistance to *B. cinerea*. The increased susceptibility of the *cdk8* mutants to *A. brassicicola* is consistent with the positive role of CDK8 in the transcriptional activation of defensin gene *PDF1.2* and in JA-mediated defense [87]. It is also consistent with the association of CDK8 with the promoter of AGMATINE COUMAROYL TRANSFERASE to promote its transcription and the biosynthesis of the defense-active secondary metabolites hydroxycinnamic acid amides [87]. The negative role of CDK8 in the resistance to *B. cinerea* is related to its role in cuticle development through interaction with the transcription factor WAX INDUCER1 (WIN1) [87]. Impaired defense responses in the *cdk8* mutant are masked by its altered cuticle, which results in increased resistance, specifically to *B. cinereal* [87].

## 5. Mediator Complex as a Central Regulator of ABA-Mediated Stress Responses

Abscisic acid (ABA) is a key stress-signaling phytohormone that accumulates under osmotic stress conditions, such as drought and high salinity, and plays an important role in plant stress responses [88]. Core components in ABA signaling include ABA receptors PYRABACTIN RESISTANCE1 (PYR1)/PYR1-LIKE (PYL)/REGULATORY COMPONENTS OF ABA RECEPTORS (RCAR), protein phosphatase 2C (PP2C) and sucrose non-fermenting 1-related protein kinase 2 (SnRK2) kinases [88]. In the absence of ABA, PP2Cs interact with SnRK2s and prevent the activation of SnRK2s. The inactive SnRK2s are unable to phosphorylate downstream substrates to activate ABA response. In the presence of ABA, PYR/PYL/RCAR receptors bind ABA and interact with PP2Cs to release SnRK2s, which are then activated by autophosphorylation of the activation loop and phosphorylate downstream substrate proteins to activate ABA response. A Raf-like kinase can also activate SnRK2 by phosphorylating the activation loop [89,90,91]. Downstream substrates of SnRK2s include genes encoding transcription factors such as ABI5 (ABA-INSENSITIVE 5), RAP2.6 (related to AP2.6) and AREBs (ABA-responsive element binding proteins), which regulate the transcription of genes that contain ABA-responsive elements (ABREs) in their promoters [88] (Figure 4).

Genetic analysis indicates that plant Mediator subunit MED25, which functions as a positive regulator of JA signaling through interaction with MYC2, serves as a negative regulator of ABA signaling and response through interaction with ABI5 [50]. In the absence of ABA, there is a low level of ABI5 but a high level of MED25 at the ABI5 target gene promoters to keep ABI5-regulated gene expression at low levels. ABA stimulates recruitment of ABI5 but reduces recruitment of MED25 to the ABI5 target gene promoter regions, thereby promoting ABI5-regulated gene expression (Figure 4). Interestingly, MED25 also positively regulates ABA-induced *ABI5* expression at the transcription level but negatively regulates ABI5 expression at the post-transcription level (Figure 4). It has been suggested that MED25 may regulate ABI5 protein stability through ABI5 phosphorylation by associated CKM of the Mediator complex [50] (Figure 4). Thus, the roles of MED25 in the regulation of ABI5 itself and ABI5-mediated ABA response are complex.

Unlike MED25, Arabidopsis MED16 and MED18 function as positive regulators of ABA response. The Mediator tail module subunit MED16 was found to be associated with MED25 in both the absence and presence of ABA [92]. Like MED25, MED16 also interacts with ABI5 and may positively regulate ABI5-mediated gene expression and ABA signaling by competing with MED25 for interaction with ABI5 [92] (Figure 4). MED18, on the other hand, directly interacts with ABI4 [31], an ERF/AP2 family transcription activator of its own gene and *ABI5* [93] (Figure 4). ABA-induced expression of ABI4 and ABI5 are reduced in *med18* mutants relative to that in WT, indicating that MED18 positively regulates the transcription of ABI4 and ABI5 [31]. Like *abi4* and *abi5* mutants, *med18* mutants are less sensitive to ABA than WT during seed germination and early growth stages. ChIP-qPCR further revealed that MED18 is recruited to the ABI4 binding site at the ABI5 promoter with or with ABA treatment [31] (Figure 4). These findings indicate that ABI4 recruits MED18 through direct interaction with *ABI5* and perhaps *ABI4* loci, in addition to activating transcription of the transcription factor genes to promote ABA-responsive gene expression and ABA response [31].

CDK8 also functions as a positive regulator of ABA signaling and drought response pathways in Arabidopsis. When compared with the wild type, *cdk8* mutants have reduced sensitivity to ABA and impaired stomatal apertures and are hypersensitive to drought stress [94]. RNA- and ChIP-seq analyses indicated that CDK8 positively regulates the expression of ABA-responsive genes [94]. Both CDK8 and SnRK2.6 interact with RAP2.6, an ERF/AP2 transcription factor that is a substrate of SnRK2.6 in ABA signaling and response [94]. RAP2.6 directly binds to the promoters of ABA-responsive genes such as RD29A and COLD-REGULATED 15A (COR15A) with GCC or DRE elements and promotes their expression [94,95] (Figure 4). In *cdk8* mutants, importantly, ABA-induced expression of both *RAP2.6* and RAP2.6-regulated ABA-responsive genes is compromised [94]. These results indicate that CDK8 links SnRK2.6-activated RAP2.6 to Pol II to promote transcription of RAP2.6 target genes in plant response to ABA and drought signals.

## 6. Roles of the Mediator Complex in Plant Responses to Abiotic Stresses

Several Mediator complex subunits have been shown to play critical roles in plant response to extreme temperature. Arabidopsis MED16, also known as SENSITIVE TO FREEZING 6 (SFR6), plays an important role in cold acclimation to survive subsequent freezing temperatures [96]. The *med16*/*sfr6* mutants are unable to express COLD ON-REGULATED (COR) genes to high enough levels in response to low, nonfreezing temperatures for cold acclimation [97,98]. COR genes contain the C-repeat motif (CRT) and are upregulated by the action of the AP2 family C-repeat binding transcription factors (CBFs). Based on further analysis of the mutants for the Mediator subunits, it has been shown that not only MED16 but also MED2 and MED14 are required for recruitment of RNA polymerase II recruitment to cold-regulated CBF target genes [99] (Figure 5A). The three Mediator subunits are also required for transcription of CBF target genes and cold acclimation-induced freezing tolerance [99] (Figure 5A). These three Mediator subunits are also required for activation of ABRE-mediated transcription in Arabidopsis [100]. Arabidopsis MED14 and MED17, on the other hand, are involved in the transcriptional regulation of plant responses to high temperature [101]. Mutants for Arabidopsis MED14 and MED17 are significantly reduced in terms of thermotolerance and in survival rate in the acquired thermotolerance when compared with WT. RNA-seq analysis revealed that a large percentage of the heat-stress-inducible genes are downregulated in the mutants when compared to WT [101]. Heat stress transcription factors (HSFs) constitute critical components of plant heat stress signal transduction, mediating the activation of heat-stress-induced genes in response to heat stress [102,103]. ChIP analysis showed that Mediator is recruited by HsfA1s, the master regulators of heat stress response, to the promoters of heat stress-inducible genes and that this recruitment is an important step for their expression, particularly for the expression of genes encoding transcription factors [101] (Figure 5B).

Several studies have shown that MED16 plays a critical role in the regulation of the uptake and homeostasis of mineral nutrients, such as iron and phosphate (Pi). Under iron limitation, expression of *FERRIC REDUCTASE*/*OXIDASE 2*(*FRO2*) and *IRON-REGULATED TRANSPORTER 1* (*IRT1*) is induced in Arabidopsis roots [104,105,106]. FRO2 reduces ferric iron to ferrous iron on the root surface, which can then be up taken into the root cells by the high affinity IRT1 iron transporter. The bHLH transcription factor FIT (FER-LIKE IRON DEFICIENCY-INDUCED TRANSCRIPTION FACTOR) is a master regulator of the expression of iron uptake genes through dimerization with the four Ib bHLH transcription factors (bHLH38, 39, 100 and 101) [107,108,109]. MED16 interacts with FIT and enhances the binding of the FIT/bHLH complexes to *FRO2* and *IRT1* promoters under iron limitation, indicating a critical role of the Mediator subunit in FIT-mediated expression of the iron-assimilating genes [110] (Figure 5C). Indeed, expression of many of FIT-mediated iron-deficient response genes, including *FRO2* and *IRT1*, is compromised in the *med16* mutants. Furthermore, the *med16* mutants contain a reduced iron concentration and exhibit severe leaf chlorosis under iron limitation. MED16 also interacts with MED25, the mutants of which are also altered in response to iron deficiency, including reduced expression of *FIT*, *IRT1* and *FRO2* [111]. MED25 interacts with ETHYLENE-INSENSITIVE 3 (EIN3) and EIN3-LIKE 1 (EIL1), two transcription factors in ethylene signaling [111] (Figure 5C). It is known that ethylene promotes iron acquisition, in part through stabilization of FIT by EIN3 and EIL1, through direct physical interaction [112]. The demonstrated interactions of EIN3 and EIL1 not only with FIT but also with MED25 indicate that that these transcription factors also cooperate in the activated transcription of iron-assimilating genes (Figure 5C).

MED16 and iron are also involved in plant response to Pi deficiency [113]. Low Pi availability induces root architecture remodeling from indeterminate to determinate primary root growth, which is linked to iron uptake and accumulation through reactive oxygen species (ROS) to regulate stem cells [114]. The SENSITIVE TO PROTEIN RHIZOTOXICITY 1(STOP1) transcription factor and its target gene product, ALUMINUM-ACTIVATED MALATE TRANSPORT1 (ALMT1), are critical components of the pathway [115]. ALMT1 functions in exudation of malate, which binds iron and other cations to increase available phosphates for root uptake; and to reduce toxic Fe^3+^ and aluminum ion levels. MED16 interacts with STOP1 and is required for transcriptional activation of STOP1 targets, including *ALMT1* and associated malate exudation and other responses under low Pi [113] (Figure 5D).

ROS, such as H_2_O_2_, as important regulators of plant growth, development, stress response and cell death, can regulate expression of a broad spectrum of genes. Very recently, it was shown that in Arabidopsis, MED8 functions as a suppressor of H_2_O_2_-induced gene expression [116]. Mutation of *MED8* enhances activation of the SA pathway and accelerates cell death in a CATALASE 2 (CAT2)-deficient mutant background with constitutive oxidative stress. The *med8* seedlings displayed increased tolerance to oxidative stress generated by the herbicide methyl viologen (MV), which is associated with enhanced transcriptional activation of defense-related genes, particularly those in the SA- and JA-related pathways. Both immunoprecipitation and yeast two-hybrid screens identified a substantial number of MED8-interacting proteins, including other Mediator subunits, transcriptional repressors and proteins involved in miRNA biogenesis [116]. These results indicate that MED8 regulates ROS-induced gene expression through multiple mechanisms. Among the MED8-interacting proteins are NEGATIVE ON TATA-LESS (NOT) proteins, which are components of the CATABOLITE REPRESSION4 (CCR4)-NOT complex, a multisubunit complex and key regulator of gene expression at all stages—from transcription in the nucleus to mRNA degradation in the cytoplasm—in all eukaryotes [117] (Figure 5E). Further functional analysis with Arabidopsis NOT2 supports the supposition that MED8 associates with NOT2 and perhaps other unknown repressors, in addition to negatively regulating plant tolerance to oxidative stress by suppressing ROS-induced defense genes [116].

Plant Mediator complex is also involved in plant responses to other abiotic stresses, including salt. In a relatively recent report, it was shown that Mediator subunit MED25 regulates shade-induced hypocotyl elongation in tomato, which is an important adaptive response to the depletion of photosynthetically active light [118]. The bHLH transcription factors PHYTOCHROME-INTERACTING FACTORS (PIFs) play a critical role in the shade-avoidance response by regulating accumulation of plant hormones, such as auxin, and expression of genes associated with cell expansion [119]. PIF4 plays a key role in shade-induced hypocotyl elongation by regulating the expression of auxin biosynthetic and signaling genes [120]. Tomato MED25 physically interacts with PIF4 at the promoter regions of PIF4 target genes to recruit Pol II for induction of transcription to regulate shade-induced hypocotyl elongation [118] (Figure 5F).

## 7. Regulation of Phenylpropanoid Biosynthesis by Plant Mediator Complex

Plant phenylpropanoids are phenylalanine-derived secondary metabolites with diverse roles in growth and development, as well as responses to environmental stimuli [121]. Lignin, a plant secondary cell wall component, is synthesized from phenylpropanoids (hydroxycinnamyl alcohols or monolignols). Lignin confers not only mechanical strength and imperviousness to the cell wall but also protective functions against biotic and abiotic stress [122]. Soluble phenylpropanoids, such as hydroxycinnamate esters, flavonols and anthocyanins, play an important role in plant–environment interactions by absorbing damaging ultraviolet (UV) light, defending against pathogens and herbivores and attracting pollinators. Studies conducted in the past decade have provided extensive evidence that specific plant Mediator subunits play important roles in the regulation of the plant phenylpropanoid pathway.

Some soluble phenylpropanoids, such as sinapoyl malate, accumulate in the epidermis of Arabidopsis leaves and can be easily detected due to their UV-adsorbent and fluorescent nature [123]. Based on altered fluorescence under UV, Arabidopsis *reduced epidermal fluorescence 4* (*ref4*) mutants with dominant mutations in *REF4*/*MED5b* have been isolated [123,124]. The *ref4*/*med5* dominant mutants display phenotypes of dwarfing and reduced accumulation of phenylpropanoids [124,125]. The dominant *ref4-3* mutant protein also inhibits the ability of the MYB75 transcription factor to induce *PAL1* expression and anthocyanin accumulation [125]. On the other hand, mutations of both MED5a and its paralog MED5a enhance expression of phenylpropanoid biosynthetic genes and increase accumulation of phenylpropanoids [125]. These results indicate that the Mediator MED5 subunits function as repressors of the transcription of phenylpropanoid biosynthetic genes and play a critical role in phenylpropanoid homeostasis.

The ability of Arabidopsis MED5a and MED26 to repress the transcription of phenylpropanoid biosynthetic genes for the regulation of phenylpropanoid homeostasis has also been demonstrated by the rescue of the Arabidopsis lignin-deficient *ref8* mutants by disruption of both MED5a and MED5b [126]. Lignin polymers are synthesized by polymerization of a combination of three related but distinct hydroxycinnamyl alcohols, generating 4-hydroxyphenyl (H), guaiacyl (G) and syringyl (S) lignin subunits [127]. *REF8* encodes *p*-coumaroyl-shikimate 3′-hydroxylase (C3′H), which catalyzes a step upstream of H lignin subunit but downstream of G and S lignin subunit synthesis in the lignin biosynthetic pathway. Despite the presence of all required biosynthetic enzymes for the synthesis of H lignin subunits, *ref8*/*c3*′*h* mutants accumulate substantially reduced levels of total lignin, in addition to stunted growth and developmental phenotypes [126]. These findings indicate that the part of the lignin biosynthetic pathway upstream of the REF8/C3′H-catalyzed step is actively repressed in the *ref8*/*c3*′*h* mutants. Disruption of both MED5a and MED5b restores the growth and lignin accumulation of the *ref8*/*c3*′*h* mutants. Unlike WT, with very low H lignin, *med5a*/*5b*/*ref8* mutant plants contain almost exclusively H lignin, with little G and S lignin [126]. These results indicate that the active repression of the lignin biosynthetic pathway in response to low REF8/C3′H activity is dependent on Mediator subunits MED5a and MED5b.

Likewise, MED5a and MED5b are required for the repressed accumulation of soluble phenylpropanoids, such as hydroxycinnamate esters (HCRs) and anthocyanins, in the mutant for *FERULIC ACID HYDROXYLASE1* (*FAH1*) [128]. *FAH1* encodes FERULATE 5-HYDROXYLASE (F5H), which catalyzes the hydroxylation of coniferaldehyde and coniferyl alcohol, which are required for the subsequent formation of sinapoylated compounds and syringyl lignin [129]. Mutation of *FAH1* does not affect plant morphology or total levels of lignin, which is almost exclusively G lignin. The *fah1* mutant is also considerably reduced in the accumulation of total HCEs and anthocyanins, indicating that the biosynthetic pathways for these soluble phenylpropanoids are also actively repressed. Disruption of both MED5a and MED5b results in increased accumulation of HCEs and anthocyanins in the *fah1* mutant to close to WT levels, indicating that repression of both HCE and anthocyanin biosynthesis in *fah1* mutants is dependent on MED5a and MED5b [128].

Both forward and reverse genetic approaches have been used to identify suppressors of the Arabidopsis semidominant *med5b*/*ref4-3* mutant, which displays reduced soluble phenylpropanoid accumulation, decreased lignin levels and dwarfism. Analysis of these suppressors has provided insights into the molecular basis for the repression of phenylpropanoid biosynthesis by MED5a and MED5b. First, among the suppressors of the semidominant *med5b*/*ref4-3* mutant isolated using the forward genetic screens are mutants for MED2, MED5, MED16 and MED23, which are all tail-module subunits of the Mediator complex [130]. These suppressors restore the growth and/or phenylpropanoid levels in the *med5b*/*ref3-4* mutant background. RNA-seq analysis showed that the suppressors reverse many of the widespread gene expression changes in the *med5b*/*ref3-4* mutant, including the upregulation of genes encoding negative regulators of the phenylpropanoid pathway [130]. Second, reverse genetic analysis has also identified CDK8, a subunit of CKM, as required for growth inhibition and suppression of phenylpropanoid biosynthesis in the *med5b*/*ref3-4* mutant [131]. Using a kinase-deficient CDK8 mutant protein, it was further demonstrated that the inhibited growth of *med5b*/*ref3-4* mutant is not dependent on the kinase activity of CDK8 [131]. Intriguingly, disruption of *MED25*, which also encodes a tail-module subunit of the Mediator complex, does not suppress *med5b*/*ref3-4* [130]. The demonstrated interdependence between the *med5b*/*ref3-4* mutant protein and other Mediator subunits in the repression of phenylpropanoid biosynthesis provides strong genetic and molecular evidence for the regulation of the phenylpropanoid pathway by the Mediator complex. The differential roles of different Mediator subunits, including those belonging to the same tail module, in the regulation of phenylpropanoid homeostasis also raise important questions about the molecular basis for the functional specificity of Mediator subunits.

## 8. Summary and Prospects

Over the past 15 years or so, considerable progress has been made in the investigation of plant Mediator structure and function. Combined biochemical, molecular and bioinformatics approaches have revealed that the structure of plant Mediator complex is highly similar to that of yeast and animal Mediator complexes [13] (Table 1). Forward and reverse genetic analyses of Arabidopsis Mediator complex subunits have revealed important and broad roles of plant Mediator in growth and development and, particularly, in plant adaptive responses to both biotic and abiotic stresses (Table 1). Specific Mediator subunits have been established to play key roles in signaling of stress-related plant hormones, such as JA, ABA and SA (Table 1; Figure 2, Figure 3 and Figure 4). Mediator subunits also play important roles in transcriptional regulation of defense-related genes associated with PTI and SAR (Table 1 and Figure 3). Some Mediator subunits are key regulators of plant response to abiotic stresses, including extreme temperature, nutrient deficiency, salt, ROS and shade (Table 1 and Figure 5). Progress has also been made in the identification of gene-specific transcription factors that activate or repress transcription of their target genes through cooperation with Mediator to influence recruitment of Pol II to target gene promoters (Table 1). There is evidence that plant Mediator complex plays important roles in plant adaptive responses through regulation of gene expression beyond transcription or even through regulation of transcription by RNA polymerases other than Pol II. The increasing body of knowledge about the critical role of Mediator in plant gene expression has considerably enriched our understanding of the molecular basis of plant growth, development and stress responses.

Despite these substantial developments, there are still many important questions about plant Mediator complex and its roles, action mechanisms and regulation. First, the functional analysis of the Mediator complex in plant adaptive responses has been substantial but by no mean comprehensive. A substantial number of Mediator complex subunits remain with no reported functional analysis in plant growth, development and stress responses (Table 1). It is likely that mutations of genes for some plant Mediator complex subunits have no detectable phenotype in growth, development or stress tolerance and, therefore, their functional characterization may require additional approaches, such as generation of composite mutants, to overcome possible functional redundancy and use of multi-omics to detect alterations in transcriptomes, proteomes and metabolomes. The gene expression profiling and multi-omics approach has been successfully used for the characterization of some Arabidopsis Mediator mutants for altered metabolism and defense gene expression [14,100,132,133,134,135,136,137]. Secondly, even for some of the Mediator subunits that have been functionally analyzed with altered phenotypes, the underlying molecular mechanisms by which they are recruited to the target gene promoter and influence their transcription are still unclear (Table 1). Thirdly, the roles of the Mediator subunits that have been functionally characterized are diverse and often distinct from one another. The molecular basis for the functional specificity of Mediator subunits has been explored but requires further investigation to better understand the dynamic nature of the Mediator complex in the regulation of target genes in plants. Fourthly, it will be important to establish the roles of the Mediator complex in the steps of gene expression other than transcription and in transcription by RNA polymerases other than Pol II, which will be highly significant for a better understanding of not only the function of plant Mediator complex but also the general transcription system in plants. Finally, a majority of the published research on plant Mediator complex has been carried out in Arabidopsis; therefore, it is important to expand research into other plants, including important crop plants, to develop important knowledge for development of new strategies and targets for plant improvement.

## Figures and Tables

**Figure 1 ijms-23-06170-f001:**
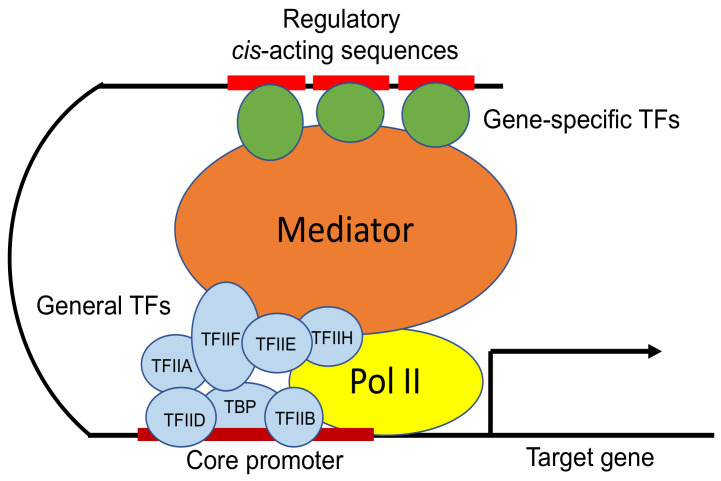
Eukaryotic transcription initiation by RNA polymerase II (Pol II). Gene-specific transcription factors (TFs) bind regulatory cis-acting sequences and, through DNA looping, also interact with the Mediator complex to coordinate the assembly of PIC, consisting of Pol II and general TFs at the core promoters, to facilitate target gene transcription.

**Figure 2 ijms-23-06170-f002:**
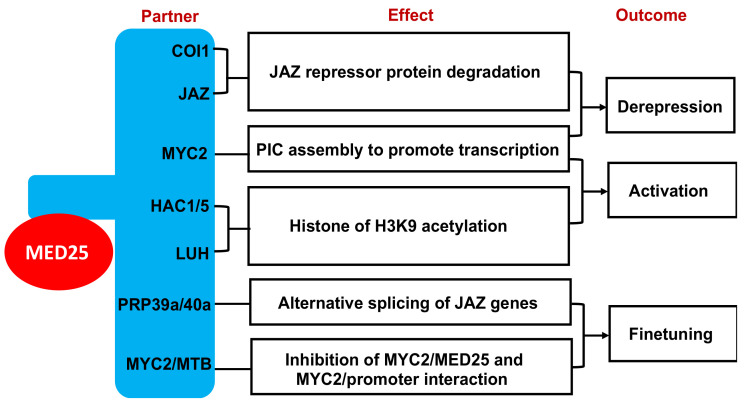
Mediator subunit MED25 coordinates with various partners in the derepression, activation and finetuning of transcription of JA-responsive genes.

**Figure 3 ijms-23-06170-f003:**
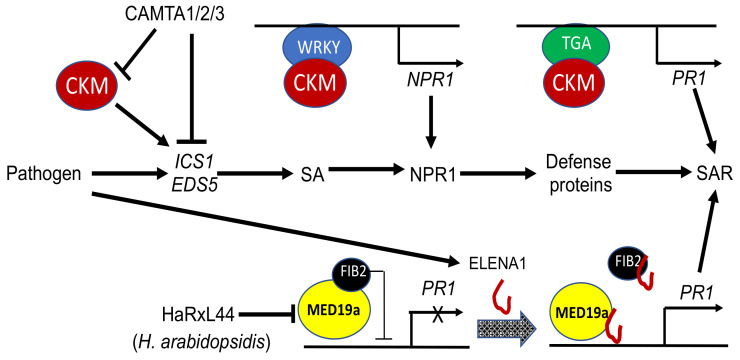
Roles of Mediator subunits in plant defense. Whereas CAMTA1/2/3 transcription factors suppress SA accumulation, CKM is a positive regulator of SA biosynthetic *ICS1* and *EDS5* genes to promote SA accumulation. CMK also coordinates with WRKY and TGA transcription factors to promote *NPR1* and *PR1* expression, respectively. The Mediator subunit MED19a is also involved in the regulation of pathogen-induced PR1 expression. In the absence of pathogen infection, MED19a is associated with MED36a/FIB2, which suppresses *PR1* expression. Pathogen infection induces ELENA1, a long non-coding RNA, which binds both MED19a and MED36a/FIB2 to disrupt their association, thereby activating *PR1* expression. The nuclear HaRx44 effector from oomycete downy mildew pathogen *H. arabidopsidis* interacts with MED19a and targets its degradation by the proteasome system.

**Figure 4 ijms-23-06170-f004:**
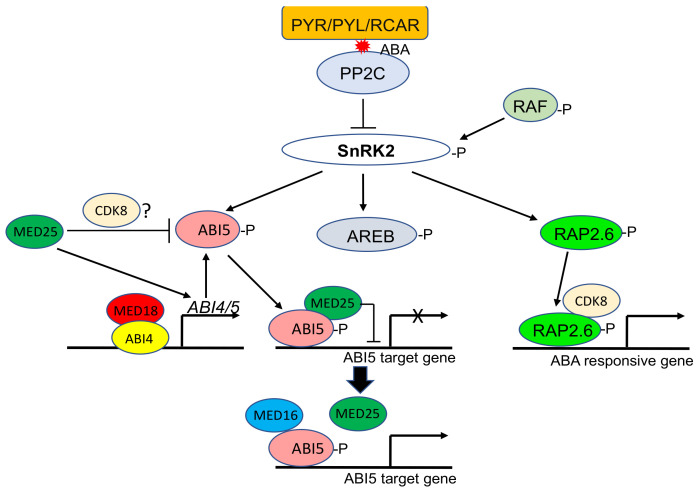
Roles of Mediator subunits in ABA signaling and response. Binding of ABA by the PYR/PYL/RCAR receptors promote their interaction with PP2C to release SnRK2s, which can activate downstream ABI5, AREB and RAP2.6 transcription factors through phosphorylation to activate ABA-responsive genes. SnRKs can also be activated through phosphorylation by RAF kinases. MED18 can interact with ABI4 to promote transcription of *ABI4* and *ABI5*. MED25 interacts with ABI5 to suppress expression of ABI5 target genes. MED16 competes with MED25 for binding to ABI5 to activate ABI5 target genes. MED25 also positively regulates transcription of *ABI5* but negatively regulates ABI5 protein stability, possibly through CDK8. CDK8 also interacts with RAP2.6 to positively regulate expression of ABA-responsive genes.

**Figure 5 ijms-23-06170-f005:**
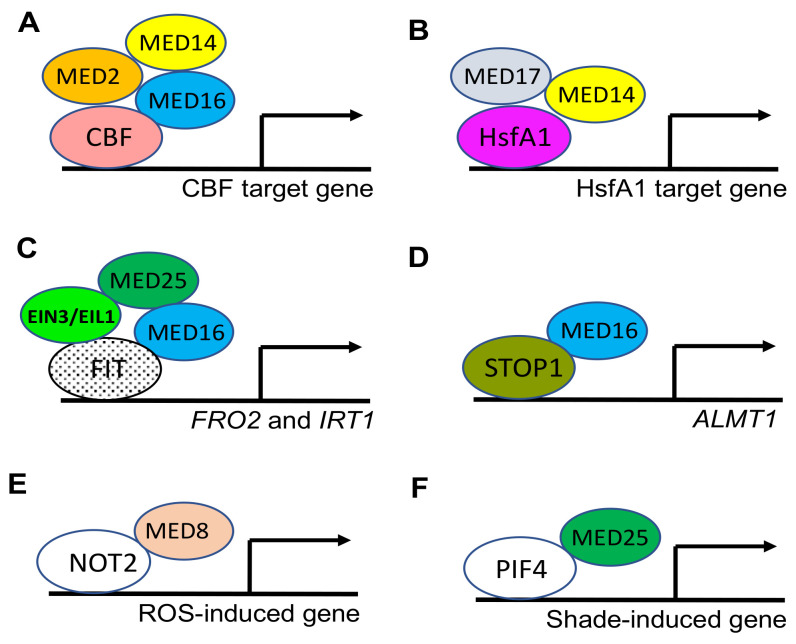
Mediator subunits coordinate with gene-specific transcription factors to promote transcription of genes associated with plant responses to cold (**A**), heat (**B**), deficiency of iron (**C**) and phosphate (**D**), ROS (**E**) and shade (**F**).

**Table 1 ijms-23-06170-t001:** Arabidopsis Mediator subunits and their roles in plant adaptive responses.

Moule	Subunit	Gene Identifier	Homolog ^1^		Roles in Plant Adaptive Responses ^2^	
			Yeast	Human	Interacting Partners	Regulated Processes
Head	MED6	At3g21350	+	+		
	MED8	At2g03070	+	+	MED25, FAMA, NOT2	Plant defense, ROS response
	MED11	At3g01435	+	+		
	MED17	At5g20170	+	+	HsfA1	Heat response
	MED18	At2g22370	+	+	NRPD2a, YY1, HLS1, ABI4	Plant defense, ABA signaling
	MED19a	At5g12230	+	+	ELENA1, HaRxL44	Plant–pathogen interactions
	MED19b	At5g19480	+	+	Unknown	Unknown
	MED20a	At2g28230	+	+	Unknown	Unknown
	MED20b	At4g09070	+	+	Unknown	Unknown
	MED20c	At2g28020	+	+	Unknown	Unknown
	MED22a	At1g16430	+	+	Unknown	Unknown
	MED22b	At1g07950	+	+	Unknown	Unknown
	MED28	At3g52860	−	+	Unknown	Unknown
	MED30	At5g63480	−	+	Unknown	Unknown
Middle	MED1	At2g15890	+	+	Unknown	Unknown
	MED4	At5g02850	+	+	Unknown	Unknown
	MED7a	At5g03220	+	+	Unknown	Unknown
	MED7b	At5g03500	+	+	Unknown	Unknown
	MED9	At1g55080	+	+	Unknown	Unknown
	MED10a	At5g41910	+	+	TPL	JA signaling
	MED10b	At1g26665	+	+	Unknown	Unknown
	MED14	At3g04740	+	+	MED2. MED16, HsfA1	Cold response, heat response, plant defense
	MED21	At4g04780	+	+	TPL, HUB1	Plant defense
	MED26a	At3g10820	−	+	Unknown	Phenylpropanoid biosynthesis
	MED26b	At5g05140	−	+	Unknown	Unknown
	MED26c	At5g09850	−	+	Unknown	Unknown
	MED31	At5g19910	+	+	Unknown	Unknown
Tail	MED2	At1g11760	+	+	CBFs	Cold response, phenylpropanoid biosynthesis
	MED3	At3g09180	+	+	Unknown	Unknown
	MED5a	At3g23590	+	+	Unknown	Phenylpropanoid biosynthesis
	MED5b	At2g48110	+	+	Unknown	Phenylpropanoid biosynthesis
	MED15a	At1g15780	+	+	Unknown	Plant defense
	MED15b	At1g15770	+	+	Unknown	Plant defense
	MED15c	At2g10440	+	+	Unknown	Plant defense
	MED16	At4g04920	+	+	MED25, ABI5, CBFs, FIT, STOP1	ABA signaling, cold responses, response to Fe and Pi limitation, phenylpropanoid biosynthesis
	MED23	At1g23230	−	+	Unknown	Phenylpropanoid biosynthesis
	MED25	At1g25540	−	+	COI1, JAZ, ORA59, ERF1, MYC2, HAC1, PRP39a, PRP40a, ABI5, EIN3/EIL1, PIF4	JA signaling, plant defense, ABA signaling, ethylene signaling, shade response
CKM	MED12	At4g00450	+	+	WRKY6, WRKY18. TGAs	SAR
	MED13	At1g55325	+	+	TPL	SAR
	CDK8	At5g63610	+	+	WIN1, RAP2.6	SAR, plant defense, ABA signaling, phenylpropanoid biosynthesis
	CYCCa	At5g48630	+	+	Unknown	Unknown
	CYCCb	At5g48640	+	+	Unknown	Unknown
Unknown	MED34	At1g31360	−	−	Unknown	Unknown
	MED35a	At1g44910	−	−	Unknown	Unknown
	MED35b	At3g19670	−	−	Unknown	Unknown
	MED35c	At3g19840	−	−	Unknown	Unknown
	MED36a	At4g25630	−	−	ELENA1	SAR
	MED36b	At5g52470	−	−	Unknown	Unknown
	MED37a	At5g28540	−	−	Unknown	Unknown
	MED37b	At1g09080	−	−	Unknown	Unknown
	MED37c	At3g12580	−	−	Unknown	Unknown
	MED37d	At5g02500	−	−	Unknown	Unknown
	MED37e	At5g42020	−	−	Unknown	Unknown
	HAC1	At1g79000	−	−	MED25	JA signaling
	HAC5	At3g12980	−	−	MED25	JA signaling

^1^ + indicates that the organism contains the homolog known as a Mediator subunit; − indicates that the organisms does not contain the homolog known as a Mediator subunit. ^2^ The interacting partners and regulated processes of only those Mediator subunits that have been analyzed to have critical roles in plant adaptive responses to environmental stresses are indicated and are also discussed in the review.

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
