# Peer review of "The Mediator Complex: A Central Coordinator of Plant Adaptive Responses to Environmental Stresses"

_ijms, 2022, doi:10.3390/ijms23116170_

Round 1
Reviewer 1 Report
The ms by Chen et al is a well written review, very clear, giving a very detailed description of the panorama on the mechanisms of regulation of the transcription factor action through the Mediator complex in plants. The authors did a very good work writing this ms and only minor spell checking and corrections are required.
Regarding this the authors should review their ms and avoid repetitions/errors like those indicated below:
on line 105 "could have implication implications...", the repetition should be removed;
on line 135-136 "CYCLIC ADENOSINE 135 MONOPHOSPHATE RESPONSE ELEMENT BING PROEIN (CREB) BING PROEIN" where I presume they wanted to write BINDING PROTEIN (and again there seems to be a repetition);
on line198 "...with components JA signaling..." should be "...components of JA signaling...";
line 225 "for fine-turning" should likely be "for fine tuning".
Otherwise I would like to compliment the authors on the work done.
Author Response
The ms by Chen et al is a well written review, very clear, giving a very detailed description of the panorama on the mechanisms of regulation of the transcription factor action through the Mediator complex in plants. The authors did a very good work writing this ms and only minor spell checking and corrections are required.
Response---We appreciate the reviewer’s very positive comments.
Regarding this the authors should review their ms and avoid repetitions/errors like those indicated below:
on line 105 "could have implication implications...", the repetition should be removed;
Response—We thank the reviewer’s very careful editing. We have corrected the error. The first “implication” was supposed to be “important”.
on line 135-136 "CYCLIC ADENOSINE MONOPHOSPHATE RESPONSE ELEMENT BING PROEIN (CREB) BING PROEIN" where I presume they wanted to write BINDING PROTEIN (and again there seems to be a repetition);
Response—Actually there is supposed to be a repetition of BING PROEIN. The first BINDING PROTEIN is for CREB (CYCLIC ADENOSINE MONOPHOSPHATE RESPONSE ELEMENT BING PROEIN); the second BINDING PROTEIN is for CBP (CREB-BINDING PROTEIN).
on line198 "...with components JA signaling..." should be "...components of JA signaling...";
Response—We have corrected the typo.
line 225 "for fine-turning" should likely be "for fine tuning".
Response—We have made the change as suggested by the reviewer.
Otherwise I would like to compliment the authors on the work done.
Response—Again we thank the reviewer for the positive comment.
Reviewer 2 Report
The review paper "The Mediator Complex: A Central Coordinator of Plant Adaptive Responses to Environmental Stresses" by Jieluo Chen, Su Yang, Baofang Fan, Cheng Zhu and Zhixiang Chen examines the role of specific organism formations in color cells. Gene expression occurs in the cell nucleus. The data on the literature sources devoted to the studies of these subcompartments under various kinds of stressful influences are presented in detail. Such an analysis is of particular importance for special issues of analysis and understanding of the regulation of stress signaling genes and various types of responses.
Unfortunately, the structure and a number of important points, in my opinion, are ignored in this review. Meanwhile, such an important interdisciplinary discussion would expand the accessibility and understanding for the readers of the review.
To accept the review for publication, a number of significant shortcomings should be eliminated and a number of text fragments should be moved to the appropriate sections, it is also necessary to take into account the importance of considering the structure of the described complexes and their classification, best in tabular form. Since the review should be easy to read and accessible for use.
Important notes:
The introduction does not consider aspects of the evolutionary origin of these complexes. There is no mention of the biochemical composition and features of the structure and main components. There is no comparison and discussion about the presence of differences in these complexes in plants.
Instead, data are presented on the similarities of neurotransmitter complexes in yeast (a unicellular organism) and indeterminate animals (which requires reading another article). What similarities and differences are not mentioned - this should be expanded. Discussion of the need for structural formations and remodeling of chromatin, the phase in which the complex is attached to DNA and the zone and state of chromatin in which this may not be understood. Meanwhile, the phases of the cell cycle are extremely sensitive to stress and the number of cells in different phases is not the same. In which tissue cells can such studies be carried out and why is it difficult to study. In the introduction it is not clear what the authors wanted to show in figure 1. Reading it is ambiguous. If we assume that there is one transcription factor, then what do the numerous various circles and ovals mean, if there are several, then why the authors did not need an explanation, moreover, the transcription factor marked with an arrow does not interact with either the promoter or the mediator. By the way, any physicochemical aspect of the interaction between the components of the complex is also not discussed, and this is important, since its creation requires a certain pH and the presence of certain ions, right?
Figures in the articles of such a reputable journal should be clear, unambiguous and informative, as well as their captions, since figures from open access journals are often used by teachers. I think that this figure should be carefully worked out and supplemented, including in the captions and in the description part.
Mediator modules should be described in detail as a complex, taking into account the specifics of interaction. In addition, the functions of its parts and interactions remain unreflected.
The purpose of the review, in my opinion, is not clearly expressed.
The reading of the second section again consists of an analysis of hard-to-compare data. Is it legal to put the data of a unicellular organism and a multicellular one? This is especially strange in the aspect of flowering, when in Arabidopsis this will require a radical restructuring of differentiation, which is in principle impossible in yeast.
The scheme and classification of the Mediator Complex plants would perhaps solve this problem.
Section 3. Epigenetic regulation of expression is described as some kind of expression activation. There is another problem with the drawing. If he himself is relatively clear, then the signatures do not contain decoding of abbreviations. There is also no description why this MED25 example was chosen, which means 25 cannot be understood, since there is no table with a description.
As for figure 3, the question arises whether this scheme was published earlier, is it a modification of the previously published one, or is this discovery by the authors based on the literature. This should be cleared up to avoid misunderstanding and plagiarism.
Similar remarks apply to figures elsewhere in the manuscript.
The conclusion should also be revised.
It should be concise and not have signs of discussion.
It is better to rename the existing conclusion to a section and write a new one.
I recommend supplementing the sections with tables and correcting and supplementing figures and signatures.
In its present form, the work requires changes and cannot be published without correction.
I hope the authors will be able to quickly eliminate these comments and shortcomings and the article can be considered and accepted for publication.
Author Response
The review paper "The Mediator Complex: A Central Coordinator of Plant Adaptive Responses to Environmental Stresses" by Jieluo Chen, Su Yang, Baofang Fan, Cheng Zhu and Zhixiang Chen examines the role of specific organism formations in color cells. Gene expression occurs in the cell nucleus. The data on the literature sources devoted to the studies of these subcompartments under various kinds of stressful influences are presented in detail. Such an analysis is of particular importance for special issues of analysis and understanding of the regulation of stress signaling genes and various types of responses.
Unfortunately, the structure and a number of important points, in my opinion, are ignored in this review. Meanwhile, such an important interdisciplinary discussion would expand the accessibility and understanding for the readers of the review.
To accept the review for publication, a number of significant shortcomings should be eliminated and a number of text fragments should be moved to the appropriate sections, it is also necessary to take into account the importance of considering the structure of the described complexes and their classification, best in tabular form. Since the review should be easy to read and accessible for use.
Important notes:
Response—We appreciate the reviewer’s constructive comments. In the revised manuscript, we have substantially revised the manuscript to address the reviewer’s concerns and suggestions. With these revisions, the manuscript has been substantially improved. Again, we thank the reviewer for the thoughtful comments.
The introduction does not consider aspects of the evolutionary origin of these complexes. There is no mention of the biochemical composition and features of the structure and main components. There is no comparison and discussion about the presence of differences in these complexes in plants.
Instead, data are presented on the similarities of neurotransmitter complexes in yeast (a unicellular organism) and indeterminate animals (which requires reading another article). What similarities and differences are not mentioned - this should be expanded. Discussion of the need for structural formations and remodeling of chromatin, the phase in which the complex is attached to DNA and the zone and state of chromatin in which this may not be understood. Meanwhile, the phases of the cell cycle are extremely sensitive to stress and the number of cells in different phases is not the same. In which tissue cells can such studies be carried out and why is it difficult to study.
Response—In the revised manuscript, we have added a table as suggested by the reviewer that list all identified mediator complex subunits and their modules. In the table, we also indicate those Mediator subunits that are conserved in yeast and human and those subunits that are unique in plants. In plants, comprehensive analysis of the Mediator subunit structures and functions has been mostly conducted only in Arabidopsis. With the genome sequences available from other plants, the similarity of the Mediator subunits has been previously analyzed and reported. In the revised manuscript, we have pointed out these previously published studies on the structural similarity of plant Mediator subunits. We, however, focus on this review primarily on those recent studies that provide important insights into the function of plant Mediator in plant adaptive responses.
In the introduction it is not clear what the authors wanted to show in figure 1. Reading it is ambiguous. If we assume that there is one transcription factor, then what do the numerous various circles and ovals mean, if there are several, then why the authors did not need an explanation, moreover, the transcription factor marked with an arrow does not interact with either the promoter or the mediator.
Response—We have revised Figure 1 with added description and explanation.
By the way, any physicochemical aspect of the interaction between the components of the complex is also not discussed, and this is important, since its creation requires a certain pH and the presence of certain ions, right?
Response—As we indicated earlier, this review is primarily focused on the functional aspects of plant Mediator subunits in plant adaptive responses. There are studies on the physicochemical aspects of Mediator subunits but mostly from yeast and mammalian systems and a detailed discussion on the physicochemical interaction of the Mediator subunits is beyond the scope of the review.
Figures in the articles of such a reputable journal should be clear, unambiguous and informative, as well as their captions, since figures from open access journals are often used by teachers. I think that this figure should be carefully worked out and supplemented, including in the captions and in the description part.
Response—We have revised Figure 1 and provided better description and explanation.
Mediator modules should be described in detail as a complex, taking into account the specifics of interaction. In addition, the functions of its parts and interactions remain unreflected.
The purpose of the review, in my opinion, is not clearly expressed.
Response—As indicated earlier, we have added a table in the revised manuscript, which includes identified Mediator subunits and their modules. In the table, we have also included information of interacting proteins and regulated biological processes of those Mediator subunits involved in plant adaptive responses.
The reading of the second section again consists of an analysis of hard-to-compare data. Is it legal to put the data of a unicellular organism and a multicellular one? This is especially strange in the aspect of flowering, when in Arabidopsis this will require a radical restructuring of differentiation, which is in principle impossible in yeast. The scheme and classification of the Mediator Complex plants would perhaps solve this problem.
Response—To be honest, we are not quite sure about the concerns and have tried our best to address them. We discussed Mediator subunits from yeast mainly to illustrate the conserved nature of the Mediator complex despite differences in growth, development and adaptive responses among different eukaryotes. To follow the reviewer’s suggestion, we have added a table of the Mediator subunits in the revised manuscript.
Section 3. Epigenetic regulation of expression is described as some kind of expression activation. There is another problem with the drawing. If he himself is relatively clear, then the signatures do not contain decoding of abbreviations. There is also no description why this MED25 example was chosen, which means 25 cannot be understood, since there is no table with a description.
Response—The full description of H3K9 acetylation signature (histone H3 lysine 9) has been provided in the text of the revised manuscript. MED25 is discussed in this section because of its crucial role in JA signaling. We have also provided additional description of MED25 in Table 1, in the revised manuscript.
As for figure 3, the question arises whether this scheme was published earlier, is it a modification of the previously published one, or is this discovery by the authors based on the literature. This should be cleared up to avoid misunderstanding and plagiarism.
Similar remarks apply to figures elsewhere in the manuscript.
Response—All the figures in the manuscripts are original so there is no issue of plagiarism.
The conclusion should also be revised.
It should be concise and not have signs of discussion.
It is better to rename the existing conclusion to a section and write a new one.
I recommend supplementing the sections with tables and correcting and supplementing figures and signatures.
In its present form, the work requires changes and cannot be published without correction.
I hope the authors will be able to quickly eliminate these comments and shortcomings and the article can be considered and accepted for publication.
Response—In “Summary and perspectives” we have summarized the advances in the areas of plant mediators in plant responses to environmental stresses. We have also pointed out the remaining questions to be addressed in the future. We think that to be valuable for future research, a review needs to provide discussion on not only the past research work but also future research directions that could lead to a better understanding of the important scientific subject being reviewed.
Round 2
Reviewer 2 Report
The manuscript "The Mediator Complex: A Central Coordinator of Plant Adaptive Responses to Environmental Stresses" by Jieluo Chen, Su Yang, Baofang Fan , Cheng Zhu, Zhixiang Chen has been significantly revised and can be accepted for publication.